# Antioxidant Effects of Turmeric Leaf Extract against Hydrogen Peroxide-Induced Oxidative Stress In Vitro in Vero Cells and In Vivo in Zebrafish

**DOI:** 10.3390/antiox10010112

**Published:** 2021-01-14

**Authors:** Sera Kim, Mingyeong Kim, Min-Cheol Kang, Hyun Hee L. Lee, Chi Heung Cho, Inwook Choi, Yongkon Park, Sang-Hoon Lee

**Affiliations:** 1Korea Food Research Institute, Wanju 55365, Korea; K.sera@kfri.re.kr (S.K.); mckang@kfri.re.kr (M.-C.K.); hyunheelee@kfri.re.kr (H.H.L.L.); Chiheungcho@kfri.re.kr (C.H.C.); choiw@kfri.re.kr (I.C.); ykpark@kfri.re.kr (Y.P.); 2Department of Food Biotechnology, Korea University of Science and Technology (UST), Daejeon 34113, Korea; 50034@kfri.re.kr

**Keywords:** turmeric leaves, antioxidants, reactive oxygen species, zebrafish

## Abstract

Oxidative stress, caused by the excessive production of reactive oxygen species (ROS), results in cellular damage. Therefore, functional materials with antioxidant properties are necessary to maintain redox balance. Turmeric leaves (*Curcuma longa* L. leaves; TL) are known to have antioxidant properties, including 2,2-Diphenyl-1-picrylhydrazyl (DPPH), 2,2′-Azino-di-(3-ethylbenzothiazoline)-6-sulfonic acid (ABTS), and Hydrogen peroxide (H_2_O_2_) radical scavenging activity in several studies. The antioxidant effects of TL come from distinct bioactive compounds, such as curcumin, total phenolic compounds, and flavonoids. Therefore, in this study, the antioxidant effects of a water extract of TL (TLE) against H_2_O_2_ treatment were assessed in vitro Vero cells and in vivo zebrafish models. The intracellular ROS generation and the proportion of sub-G1 phase cells were evaluated in H_2_O_2_- or/and TLE-treated Vero cells to measure the antioxidant activity of TLE. TLE showed outstanding intracellular ROS scavenging activity and significantly decreased the proportion of cells in the sub-G1 phase in a dose-dependent manner. Furthermore, cell death, ROS generation, and lipid peroxidation in the H_2_O_2_-treated zebrafish model were attenuated as a consequence of TLE treatment. Collectively, the results from this study suggested that TLE may be an alternative material to relieve ROS generation through its antioxidant properties or a suitable material for the application in a functional food industry.

## 1. Introduction

Reactive oxygen species (ROS) refers to a number of different reactive oxygen-containing molecules that are generated during cellular metabolism [1]. ROS causes oxidative stress which can damage essential macromolecules including proteins, DNA, and lipids, resulting in alteration of biological activity [2]. In general, the generation of ROS is determined by balance between ROS production and removal, which is called redox balance or redox homeostasis [3]. However, the ROS-antioxidant balance may be collapsed in pathological conditions after external stimuli exposure [4]. Then, ROS can be accumulated and trigger several pathological processes including inflammation, apoptosis and autophagy [5]. This redox imbalance finally resulted in various diseases including cancer [4], cardiovascular diseases [6], neurological disorders [7], and diabetes [8]. Accordingly, the supplementation of antioxidants is required to maintain redox balance and to prevent diseases arising from ROS and/or ROS imbalance-induced oxidative stress [9].

*Curcuma longa* L. (Zingiberaceae), commonly called turmeric, is an herbaceous plant and cultivated in many Asian countries including India and China [10]. Traditionally, turmeric has been used as a medicinal plant with its various biological activities such as strengthening energy, antioxidant, antibacterial, anti-inflammatory, anticancer, and wound healing [11,12]. These functional properties result from curcuminoids, the major components of turmeric, including demethoxycurcumin, bisdemethoxycurcumin, and curcumin [13]. Curcumin, a well-known yellow pigment, is a potential substance that may control oxidative stress-induced cellular damage owing to its radical scavenging activity [12,14]. Similarly, turmeric leaves are an ingredient added to various dishes in South-East Asia, as they are believed to have antioxidant properties [15]. However, turmeric leaves are mostly wasted as byproducts, except for animal feeds after harvesting [16]. According to the research paper [17], turmeric leaves also contain bioactive compounds, such as curcumin, several phenolic compounds, and flavonoids. These compounds have been known to act as an antioxidants owing to its effective radical-scavenging activity [18]. However, there are a limited number of publications detailing the functionality of turmeric leaves. In particular, the effects of turmeric leaves on ROS-induced oxidative stress remain unclear.

In this study, we aimed to evaluate the effects of turmeric leaf extract (TLE) on hydrogen peroxide (H_2_O_2_)-induced oxidative stress. To demonstrate the effects of turmeric leaf extract, the present study investigated the composition of functional compounds in turmeric leaf extract and their effects on cellular ROS generation and apoptosis in in vitro Vero cells and in vivo zebrafish model.

## 2. Materials and Methods

### 2.1. Chemicals

Turmeric leaves were obtained from the Jindoulguem Corp. (Jindo-gun, Jeollanamdo, Korea). Dulbecco’s modified Eagle medium (DMEM), fetal bovine serum (FBS), penicillin-streptomycin, Dulbecco’s phosphate buffered saline (DPBS) and trypsin-EDTA were purchased from Gibco (Burlington, ON, Canada). Hydrogen peroxide (H_2_O_2_), RNase A, 2,7-dichlorofluorescein diacetate (DCFH-DA), propidium iodide (PI), Hoechst 33342, dimethyl sulfoxide (DMSO), and diphenyl-1-pyrenylphosphine (DPPP; DJ799, Dojindo Laboratories, Kumamoto, Japan) were purchased from Sigma (St. Louis, MO, USA). 19 flavonoids (acacetin, amentoflavone, astragalin, baicalein, catechin, daidzin, dismetin, eriodictyol, genistin, genistein, genkwanin, hesperidin, hesperetin, luteolin, myricetin, narirutin, quercetin, rutin, taxifolin) and 11 flavonoids (apigenin, chrysin, diosmin, isorhamnetin, kaempferide, kaempferol, myricitrin, naringenin, naringin, neohesperidin, puerarin) were purchased from Sigma-Aldrich (St. Louis, MO, USA) and Tokyo Chemical Industry (TCI), respectively. Five flavonoids (avicularin, poncirin, procyanidin B2, Procyanidin B3, spinosin) and three flavonoids (isovitexin, miquelianin, quercitrin) were purchased from Chemfaces and HWI group, respectively. Formic acid (FA) and fluorescein were purchased from Sigma-Aldrich. LCMS-grade solvents (JT Baker, Phillipsburg, NY, USA) were used as the solvents of HPLC-MS experiments. All flavonoids were carefully weighed and diluted using HPLC-grade MeOH or DMSO or acetone (JT Baker, Phillipsburg, NY, USA). The concentration of the flavonoid standard stock was set as 4 mg/mL. Fluorescein was utilized as the internal standard for the flavonoid analysis.

### 2.2. Turmeric Leaf Extraction

The turmeric seeds were sown in April and fresh turmeric leaves were harvested in November at Jindo-gun (South in Korea). They were washed three times to eliminate impurities, and hot-air dried at 50 °C for 24 h. Then, the leaf powder was extracted with distilled water at 85 °C for 150 min with 1:25 extraction ratio as an optimal extraction condition. After filtration of the turmeric leaf extract (TLE) with 0.22 μm PVDF filter, the leaf extract was diluted with distilled water at 10 mg/mL and stored at −20 °C until use.

### 2.3. High-Performance Liquid Chromatography Mass Spectrometry (HPLC-MS)

All MS experiments were performed in negative ion mode using an AB Sciex X500R QTOF equipped with an ExionLC HPLC. ACQUITY UPLC BEH C18 column (1.8 μm, 2.1 × 100 mm) was utilized for the separation of the flavonoids in the extract. 0.1% FA water and 0.1% FA ACN were used as HPLC solvent A and B, respectively. The flow rate and the column oven temperature were set as 0.3 mL/min and 40 °C, respectively. A gradient of the two solvents was set as follows: 0–3.0 min, linear increase from 0% to 10% B; 3.0–8.0 min, linear increase from 10% to 35% B; 8.0–22.0 min, linear increase from 35% to 80% B; 22.0–23.0 min, 80% B; 23.0–23.5 min, linear decrease from 80% to 0%; 23.5–28.0 min, 0% B. We inject the sample solution of 5 μL for obtaining the single HPLC-MS data. Spray voltage, CAD gas, curtain gas, and source temperature were set as −4500 V, 7, 25, and 400 °C, respectively. Collision energy and collision energy spread were set as −35 V and 20 V, respectively. Flavonoids in the extract were identified using *m*/*z* values of precursor ion and product ions and retention factor (Rf) ratio between flavonoid and internal standard obtained from the HPLC-MS/MS experiments of the flavonoid standards. The contents of the flavonoids were estimated using the internal standard method.

### 2.4. Cell Culture

Vero cells, kidney cells from African green monkey, were grown at 37 °C in a 5% CO_2_ humidified atmosphere. The cells were cultured using DMEM supplemented with 10% heat-activated fetal bovine serum (FBS), penicillin (100 unit/mL), streptomycin (100 μg/mL), and sodium pyruvate (110 μg/mL). The cells were cultured for 2 days after they reached 80% confluence. For the experiment, the cells were seeded on a 96-well plate with 1 × 10^4^ cells/well density.

### 2.5. Assessment of Cell Viability

Vero cells were seeded on a 96-well plate with 1 × 10^4^ cells/well and incubated for 24 h. After change the medium to the serum-free medium, different concentrations of TLE (10, 25, 50 and 100 μg/mL) and 600 μM hydrogen peroxide (H_2_O_2_) were treated to each well. After 24 h, aspirate the medium and treat the MTT containing medium (100 μg/mL). After 3 h, remove the medium and add 100 μL of DMSO to the wells. Measure absorbance at 570 nm using Microplate readers (Molecular Devices, San Jose, CA, USA).

### 2.6. Intracellular ROS Generation in Vero Cells

Vero cells were seeded on a 96-well black plate with 1 × 10^4^ cells/well and incubated for 24 h. After change the medium to the serum-free medium, TLE in different concentrations (10, 25, 50 and 100 μg/mL) was pre-treated to the cells for 1 h. Then, 600 μM H_2_O_2_ was treated to the cells followed by an hour incubation. Then, 10 μg/mL of ROS-specific fluorescent dye DCFH-DA solution was added to the cells. After 30 min, fluorescence was measured at a wavelength of 485 nm (excitation)/535 nm (emission) using Microplate reader (Molecular Devices, San Jose, CA, USA).

### 2.7. Cell Cycle Analysis in Vero Cells

Cell cycle analysis was carried out to determine the proportion of sub-G1 phased cells after TLE and H_2_O_2_ treatment. The cells were seeded on a 6-well plates with 2 × 10^5^ cells/well. After 24 h, TLE in different concentrations (10, 25, 50 and 100 μg/mL) was pretreated to the cells for 30 min. Then, 600 μM H_2_O_2_ was added to the cells followed by 1-h incubation. After 6 h, centrifuge the cells and supernatant, and wash the cells with ice-cold DPBS. Then, fix the cells with 70% ethanol, and add 5 μg/mL of RNase A and 10 μg/mL of PI. Incubate the cells for 30 min on ice and analyze proportion of sun-G1 phased cells using flow cytometry (Beckman Coulter, Brea, CA, USA).

### 2.8. Nuclear Staining with Hoechst 33342 and PI

Vero cells were seeded on 24-well cell imaging plates (Eppendorf, Hamburg, Germany) with of 5 × 10^4^ cells/well. After 24 h, cells were pretreated with TLE at 10 μg/mL and 100 μg/mL concentration, and incubated for 24 h. Then, 600 μM H_2_O_2_ was added to the cells for 30 min. The culture medium was removed and the cells were stained with 1 μg/mL Hoechst 33342 for 30 min and 10 μg/mL PI for 1 h. Stained nuclear was observed with excitation at 350 nm and emission at 460 nm under a fluorescence microscope (Zeiss Axio Observer A1, ZEISS, Jena, Germany).

### 2.9. Origin and Maintenance of Zebrafish

Adult zebrafish was obtained from commercial dealer (Seoul Aquarium, Jeju, Korea). Zebrafish were kept in a 3-L acrylic tank at 28.5 ± 0.5 °C with a 14/10-h light/dark cycle. The zebrafish were fed two times a day, 6 days in a week with Tetramin flake food supplemented with brine shrimp (Artemia salina). Embryos of zebrafish were obtained by natural mating and spawning within 30 min. After spawning, the embryos were moved to a Petri dish containing 1 mg/mL methylene blue solution. After disinfection for 1.5 h, the methylene blue solution was changed to fresh embryo medium.

### 2.10. Waterborne Exposure of Embryo with or without TLE and H_2_O_2_ in Zebrafish

After 7–9 h post-fertilization (hpf), embryos were individually transferred to a 12-well plate (15 embryos per group) and maintained in medium containing TLE (50, 100, and 200 μg/mL). After 1 h of incubation, 5 mM H_2_O_2_ was added to the medium, and the embryos were incubated until 24 hpf. After counting survival rate, the surviving fish were used for the analysis.

### 2.11. Estimation of Oxidative Stress-Induced Intracellular ROS and Lipid Peroxidation as well as Cell Death in Zebrafish Embryos and Image Analysis

ROS production, lipid oxidation, and cell death in zebrafish embryos were investigated according to a method proposed by Kang et al. [19]. Dichloro-dihydro-fluorescein diacetate (DCFH-DA) (D6883, Sigma-Aldrich) was used to measure the intracellular active oxygen production in zebrafish embryos, and 1,3-Bis(diphenyl-phosphino) propane (DPPP) (DJ799, Dojindo Laboratories, Kumamoto, Japan) was used to measure lipid peroxidation. Cell death in zebrafish embryos was measured by staining with acridine orange (A8097, Sigma-Aldrich), a nucleic acid-selective dye that interacts with DNA and RNA. After pretreating zebrafish embryo samples at 3–4 h post fertilization, 5 Mm hydrogen peroxide was added to the medium, and the embryos were incubated for 24 h. Then, the embryo medium was replaced and the culture observed for two days. The samples were treated with DCFH-DA (20 μg/mL), DPPP (25 μg/mL), and acridine orange (7 μg/mL), and incubated for the appropriate reaction time for each reagent. Embryos were then washed, anesthetized, and observed under a fluorescence microscope. The intensity of fluorescence staining was quantified and plotted using the ImageJ software.

### 2.12. Statistical Analysis

All experiments were conducted in triplicates. All data were presented as means ± standard deviation. One-way ANOVA analysis was used to compare the mean values of each treatment in GraphPad prism software version 8.0. Significant differences between the means were identified by the Tuckey’s regression test. Significance was established as * *p* < 0.05, ** *p* < 0.01, and *** *p* < 0.001.

## 3. Results

### 3.1. Flavonoids in TLE

The flavonoids in TLE were analyzed with HPLC-MS method and flavonoid contents were expressed as ng flavonoid per dry weight of TLE (ng/mg). 10 flavonoids were identified in TLE as diosmetin (525.40 ± 0.23 ng/mg), quercitrin (304.94 ± 7.95 ng/mg), rutin (118.33 ± 6.72 ng/mg), miquelianin (111.38 ± 6.57 ng/mg), taxifolin (92.14 ± 5.06 ng/mg), myrictrin (63.73 ± 2.31 ng/mg), puerarin (55.30 ± 4.56 ng/mg), narirutin (25.81 ± 1.44 ng/mg), naringin (25.81 ± 1.44 ng/mg), quercetin (23.31 ± 0.44 ng/mg), among 30 flavonoids analytic samples (Figure 1).

### 3.2. Effect of TLE on Cell Viability

The effect of TLE treatment on H_2_O_2_-treated Vero cells was determined by MTT assay. As shown in Figure 2, cell viability in untreated cells was considered as 100%, and in the H_2_O_2_-treated group, cell viability was reduced to 78.4%. However, the cell viabilities in all TLE treatment groups were higher than that of H_2_O_2_-treated group and showed a viability of 86.3% compared with the untreated group. These results indicated that TLE was not cytotoxic and inhibited cell death induced by H_2_O_2_ treatment in Vero cells.

### 3.3. Effect of TLE on Intracellular ROS Generation in Vero Cells

The DCFH-DA assay was performed to evaluate the intracellular ROS scavenging activity of TLE in H_2_O_2_-treated Vero cells. After 600 μM H_2_O_2_ treatment, ROS generation was increased more than approximately 7.4 times in the H_2_O_2_-treated group compared with that of the untreated group (Figure 3). However, ROS generation in the TLE treatment groups showed a dose-dependent decrease, except for the 10 μg/mL treatment. In particular, the highest concentration of TLE (100 μg/mL) showed the strongest ROS scavenging activity, of 52.6%, compared with that of the H_2_O_2_-treated cells (100%). These results indicated that TLE treatment inhibited intracellular ROS generation in H_2_O_2_-treated Vero cells.

### 3.4. Effect of TLE on the Sub-G1 Phase Cell Population in Vero Cells

To confirm the degree of DNA fragmentation, the sub-G1 phase population was measured by flow cytometry of propidium iodide (PI)-stained cells. The population in the untreated group was 0.4%, and the population in the control group had significantly increased, to 24.7%, after H_2_O_2_ treatment. In comparison with the sub-G1 population in the control group, dose-dependent decreases were observed in the TLE treatment groups, of 19.7%, 10.6%, 9.3%, and 6.3% (Figure 4). These results showed that treatment with a high concentration of TLE effectively inhibited DNA fragmentation, as shown by the reduction in the population of cells in the sub-G1 phase.

### 3.5. Nuclear Staining with Hoechst 33342 and PI

The effects of TLE on H_2_O_2_-induced cell death, apoptosis, and nuclear DNA fragmentation were assessed via nuclear staining with Hoechst 33342 and PI. As shown in Figure 5, uninjured nuclei were clearly observed in the untreated group and H_2_O_2_-treated cells were associated with increased nuclear condensation (Hoechst 33342 staining, blue fluorescence) and cell death (PI, red fluorescence). However, 10 and 100 μg/mL TLE treatment led to significant reductions in nuclear condensation and cell death in H_2_O_2_-treated cells. These results suggested that TLE treatment was able to inhibit oxidative stress-induced apoptosis in H_2_O_2_-treated Vero cells.

### 3.6. Inhibitory Effects of TLE on H_2_O_2_-Induced Cell Death, ROS Production, and Lipid Peroxidation in Zebrafish

To confirm the effect of TLE on H_2_O_2_-induced oxidative stress, we analyzed cell death, ROS production, and lipid peroxidation in zebrafish. The protective effect of TLE against H_2_O_2_-induced cell death in zebrafish was evaluated by acridine orange fluorescence staining. The cell death percentage in H_2_O_2_-treated zebrafish was considered to be 100%. TLE treatment at 100 and 200 μg/mL significantly reduced the H_2_O_2_-induced cell death in zebrafish, to 40.0% and 16.7%, respectively (Figure 6a). The antioxidant effect of TLE on ROS generation in H_2_O_2_-treated zebrafish embryos was measured by DCFH-DA fluorescence intensity. The ROS generation in H_2_O_2_-treated zebrafish was considered to be 100%. TLE treatment at 100 and 200 μg/mL significantly decreased the ROS generation in the H_2_O_2_-treated zebrafish as 51.0% and 72.6%, respectively (Figure 6b). The antioxidant effect of TLE on lipid peroxidation generation in H_2_O_2_-treated zebrafish was measured by DPPP fluorescence intensity. The lipid peroxidation in H_2_O_2_-treated zebrafish was considered to be 100%. TLE treatment of 100 and 200 μg/mL significantly decreased lipid peroxidation in H_2_O_2_-treated zebrafish in a concentration dependent manner, to 20.9%, and 6.7%, respectively (Figure 6c). Collectively, these results demonstrated that TLE exerted a protective effect against H_2_O_2_-induced oxidative stress in the zebrafish model.

## 4. Discussion

Turmeric leaves, the aerial parts of turmeric, are considered as by-products of turmeric plant after harvesting [20]. However, several studies have reported that turmeric leaves possess pharmacological properties, including antioxidant, anti-inflammatory, antitumor, and antibacterial effects, arising from its radical scavenging activity [21]. These reports suggest that turmeric leaves, considered as a byproduct, has strong potentials to be developed into a variety of functional food materials. Therefore, our previous and current studies investigated the characteristics, functionality, and optimal extraction conditions of turmeric leaves for further research and application in food industry. We analyzed biochemical properties of turmeric leaves extract in the previous study [22]. According to our previous study results [22], the total flavonoids contents of TLE were shown as 4.78 ± 0.01 mg QCE and TLE were also shown to have DPPH, ABTS, and H_2_O_2_ radical scavenging activities as 51.10 ± 2.29%, 91.08 ± 0.15%, and 25.39 ± 2.69%, respectively. Flavonoids are large compounds occurring ubiquitously in food plants and found to be strong free radical scavengers and antioxidants [23]. The antioxidant ability of TLE is also believed to be due to its various flavonoids. In the present study, we analyzed the flavonoids in TLE with HPLC-MS method. Total 10 flavonoids were identified in TLE, and diosmetin (525.40 ± 0.23 ng/mg), quercitrin (304.94 ± 7.95 ng/mg) and rutin (118.33 ± 6.72 ng/mg) were contained as major flavonoids. Diosmetin is reported that it has therapeutic effects as anti-oxidant, anti-inflammatory, anti-bacterial, and anti-cancer properties [24,25]. Especially, diosmetin is known to alleviate inflammation by repressing the NF-_Κ_B signaling pathway in lipopolysaccharides (LPS)-induced cellular model [26] and ameliorates oxidative stress and DNA damage in cellular apoptosis [27]. Quercitrin and rutin also reported that it has anticancer, antifibrosis, anti-nephropathy, anti-inflammation and antioxidant effects [28,29]. Based on these results, we concluded that TLE can be used as a functional food material with its strong radical scavenging activities.

ROS are reactive oxygen-containing molecules, mostly radicals, including singlet oxygen (^1^O_2_), superoxide radical anion (O_2_^●−^), H_2_O_2_, hydroxyl radical (^●^OH), and the hypochlorite ion (OCl^−^) [30]. Intracellular ROS are byproducts of oxygen metabolism and highly related to cell signaling and inflammation responses. Elevated ROS under normal conditions, due to the defect in biological scavengers, are required to drive regulatory pathways, including cell signaling, apoptotic pathways, and cell and tissue growth [31]. However, the overproduction of intracellular ROS can directly and indirectly damage to nucleic acids, modification of the structure and function of cellular lipids and proteins, which finally leads to the induction of apoptosis and necrosis [32,33,34]. In this regard, oxidative stress, a consequence of the imbalance between intracellular ROS accumulation and scavenging system, plays an important role in the development of cellular damage [35,36]. Indeed, many studies have investigated the role of oxidative stress in many pathological cases of cancer, neurodegenerative diseases, and metabolic diseases, such as diabetes, hypertension, and cardiovascular diseases, through the regulation of insulin signaling, inflammatory responses, and autophagy [36,37]. Therefore, antioxidants, acting to balancing the levels of ROS, are required to avoid the onset and progression of the harmful effects on cellular molecules and nearby tissues.

This study focused on the cellular H_2_O_2_ stimulation under both in vivo and in vitro condition as a distinct ROS species. H_2_O_2_ is an important metabolite in redox signaling, redox regulation, and oxidative stress-mediated cell death, where it acts as a messenger molecule [38,39]. H_2_O_2_, a stable molecule, diffuses across the cellular membrane with a specific carrier and is converted into a highly reactive hydroxyl radical [40]. This radical induces cell death via oxidative signaling; therefore, this study used H_2_O_2_ as an intracellular stimulant for oxidative stress. To investigate the ROS scavenging activity of TLE in vitro, intracellular ROS generation in Vero cells was detected by DCFH-DA staining. The DCFH-DA stain is taken up through the cellular membrane and is converted to DCFH via cellular esterase. Then, intracellular ROS converts DCFH into the fluorescent active DCF, which can be detected with a fluorimeter [39]. As we previously mentioned, the results from this study have shown that H_2_O_2_ treatment led to a significant stimulation of intracellular ROS generation, whereas all concentrations of TLE remarkably decreased ROS in Vero cells, and a dose-dependent effect was observed. These results indicated that TLE treatment may effectively regulate the ROS production induced by H_2_O_2_ treatment owing to its prominent radical scavenging activity. Therefore, cell cycle analysis was performed by PI staining to confirm the effect of TLE on ROS scavenging and apoptosis.

As many studies have reported, ROS, including H_2_O_2_, also regulate various signaling mechanisms, dysregulation of cell cycle progression, growth, and proliferation [36]. Apoptosis, also known as programmed cell death, is a specific cell death pathway characterized by morphological changes, such as chromatin condensation, nuclear fragmentation, and the formation of apoptotic bodies [41]. The cell cycle is also associated with apoptosis; cell cycle arrest in the sub-G1 phase leads induces apoptosis via effects on various signaling molecules and regulatory factors [42]. In this study, we therefore investigated the effects of TLE on cell cycle arrest in the sub-G1 phase and the population of apoptotic bodies in H_2_O_2_-treated Vero cells. The increased proportion of sub-G1 phase cells is considered as a blockade of cell cycle kinetics, caused by DNA damage; if this damage is not repaired by the DNA repair system [43], cell death may occur. To investigate the proportion of cells in the sub-G1 phase, the cell cycle was analyzed by PI staining in the presence and absence of TLE and/or H_2_O_2_ treatment. Our results showed that the proportion of sub-G1 cells in H_2_O_2_-treated cells increased owing to DNA fragmentation and apoptosis. However, different concentrations of TLE significantly reduced the proportion of sub-G1 phase cells, illustrating the antioxidant effects of TLE.

Nuclear condensation and cell death are major features of ROS-induced cellular apoptosis. To measure the protective effects of TLE treatment on H_2_O_2_-induced apoptosis in Vero cells, nuclear condensation and cell death were examined by fluorescence microscopy of Hoechst 33342 and PI dual stained cells. Hoechst 33342 exhibits blue fluorescence, and indicates cells in early apoptosis by staining the condensed nucleus, whereas PI exhibits red fluorescence, and indicates dead cells by binding permanently to their DNA [44]. As shown in Figure 4, nuclear condensation was increased in the H_2_O_2_-induced cells compared with the untreated cells. In contrast, nuclear condensation was decreased in a dose-dependent manner in the TLE-treated groups. The proportion of cell death in the TLE-treated groups was also significantly lower than that in the H_2_O_2_-treated group, indicating that the antioxidant effects of TLE occurred through inhibition of the progression of nuclear condensation and cell death.

To confirm the results of TLE treatment in vitro, zebrafish was used as an in vivo model. Zebrafish models are a popular experimental tool in many research areas, such as biology, pharmacology, and toxicology [19]. As a species, zebrafish has many advantages, including low maintenance costs, rapid embryogenesis, transparency, size, and genomic similarity to humans, compared with other vertebrate species [45,46]. Moreover, a zebrafish model stimulated with H_2_O_2_ has been successfully used to investigate the antioxidant effects of natural products in many studies [47]. Based on these reported studies, a zebrafish model stimulated with H_2_O_2_ was selected as for the in vivo investigation of the antioxidant properties of TLE. Cell death, ROS generation, and lipid peroxidation of H_2_O_2_-treated zebrafish were measured to identify the antioxidant effects of TLE treatment in this study. The results indicated that TLE dramatically decreased H_2_O_2_-induced cell death, ROS generation, and lipid peroxidation in the zebrafish model.

Collectively, the current study results showed that the water extract of turmeric leaves has strong antioxidant activity. The in vitro and in vivo experiments performed in this study also helped to determine the effects of TLE on ROS scavenging activity and oxidative stress inhibition. To the best of our knowledge, this study has provided the first evidence to show that TLE attenuated H_2_O_2_-induced cellular damage via its own ROS scavenging properties. We believe that these findings provide supporting evidence for the use of turmeric leaves as a functional material in food industry.

## 5. Conclusions

The study results suggest that the turmeric leaf extract has a strong antioxidant effect that is exerted through the suppression of H_2_O_2_-induced ROS generation in vitro and in vivo model. Therefore, the findings of this study suggest that TLE may be a food source of antioxidants, suitable for use in functional foods or as an alternative treatment for oxidative stress-induced cellular damage.

## Figures and Tables

**Figure 1 antioxidants-10-00112-f001:**
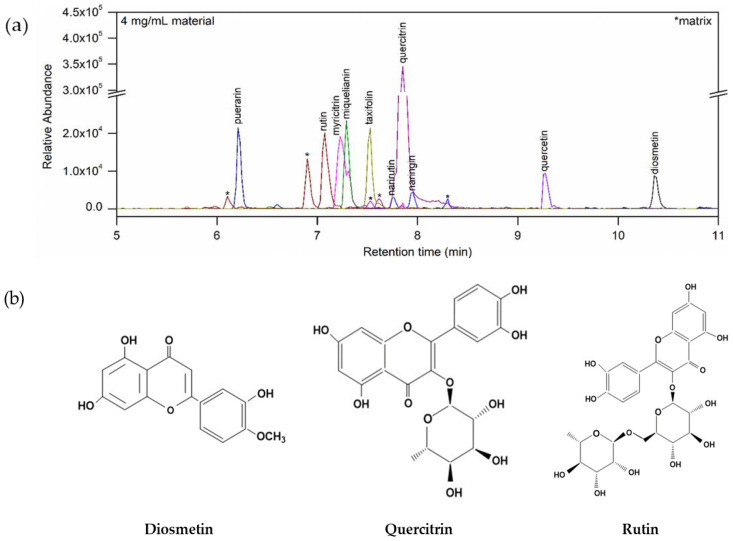
(**a**) Extracted ion chromatograms (EIC) of flavonoids and (**b**) chemical structures of three major flavonoids in the extract of 4 mg/mL.

**Figure 2 antioxidants-10-00112-f002:**
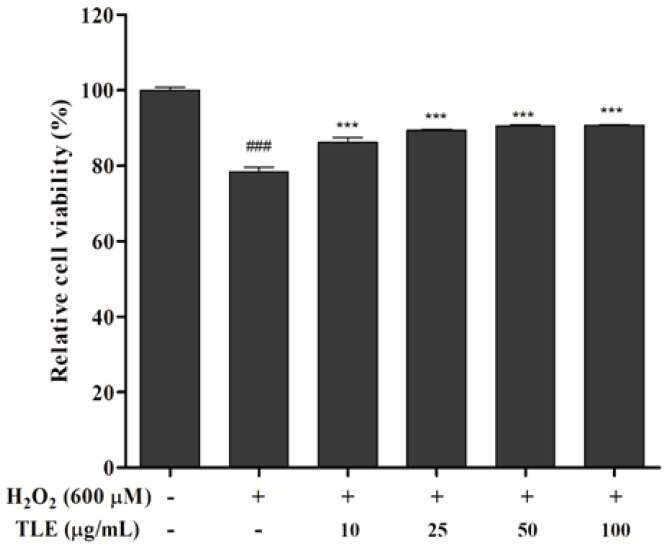
Viability of the H_2_O_2_ and/or TLE-treated Vero cells. The data were measured in triplicate and expressed as the mean ± SE. *** *p* < 0.001 compared with the H_2_O_2_-treated group and ### *p* < 0.001 compared with the untreated group.

**Figure 3 antioxidants-10-00112-f003:**
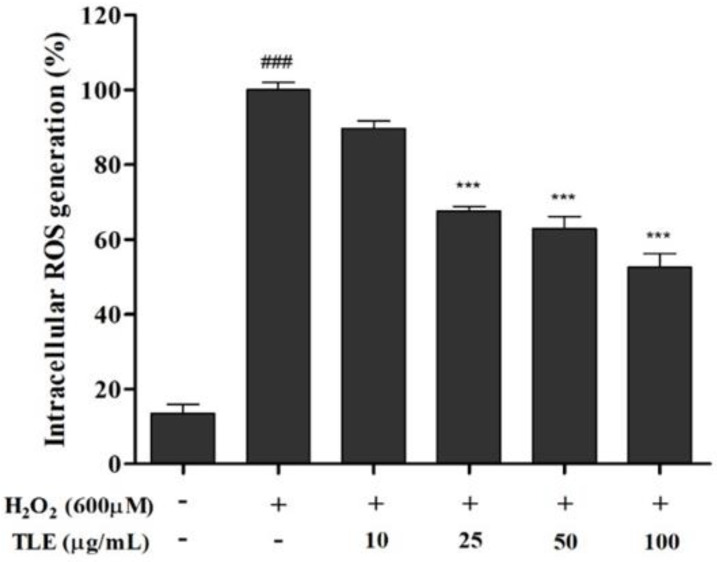
Intracellular reactive oxygen species (ROS) generation in the H_2_O_2_ and/or TLE-treated Vero cells. The data were measured in triplicate and expressed as the mean ± SE. *** *p* < 0.001 compared with the H_2_O_2_-treated group and ### *p* < 0.001 compared with the untreated group.

**Figure 4 antioxidants-10-00112-f004:**
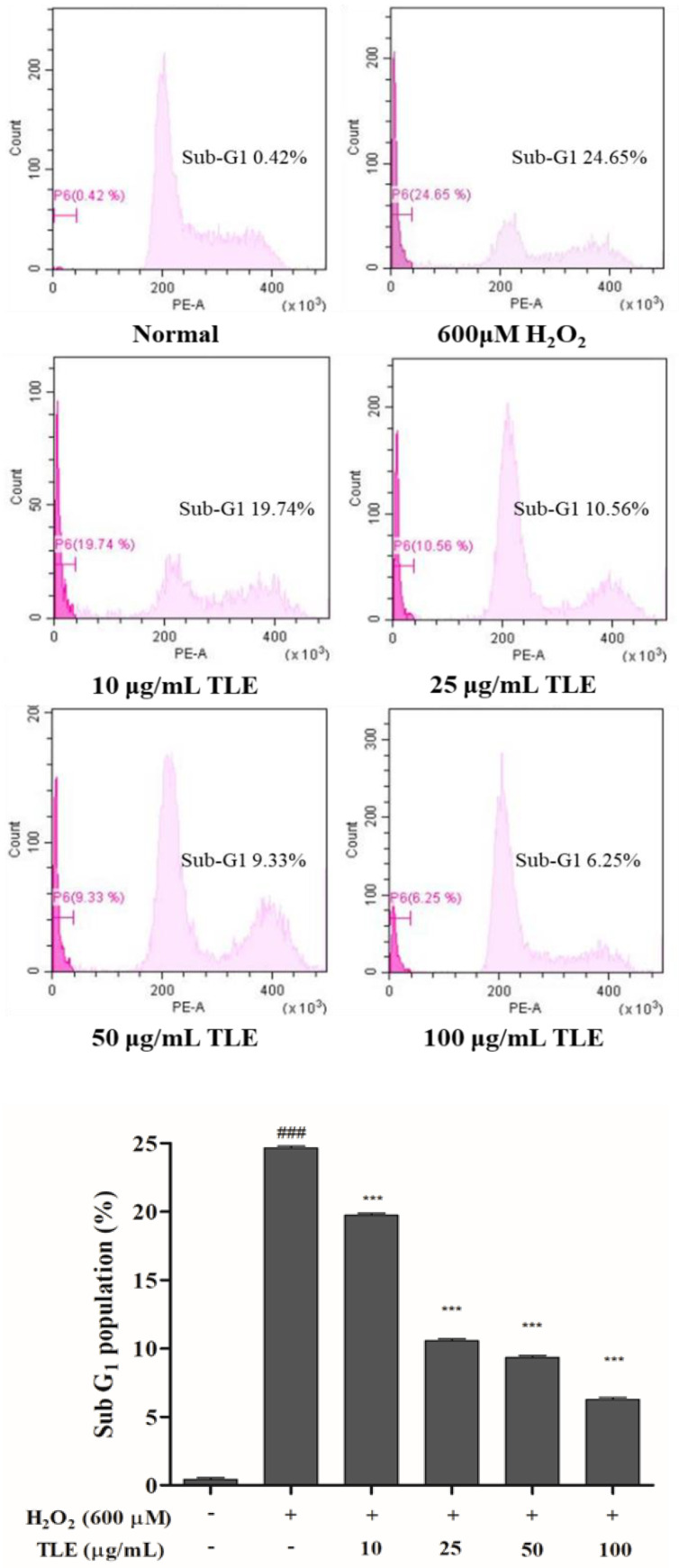
Proportion of sub-G1 phase cells in H_2_O_2_-treated and/or TLE-treated Vero cells. The data were measured in triplicate and expressed as the mean ± SE. *** *p* < 0.001 compared with the H_2_O_2_-treated group and ### *p* < 0.001 compared with the untreated group.

**Figure 5 antioxidants-10-00112-f005:**
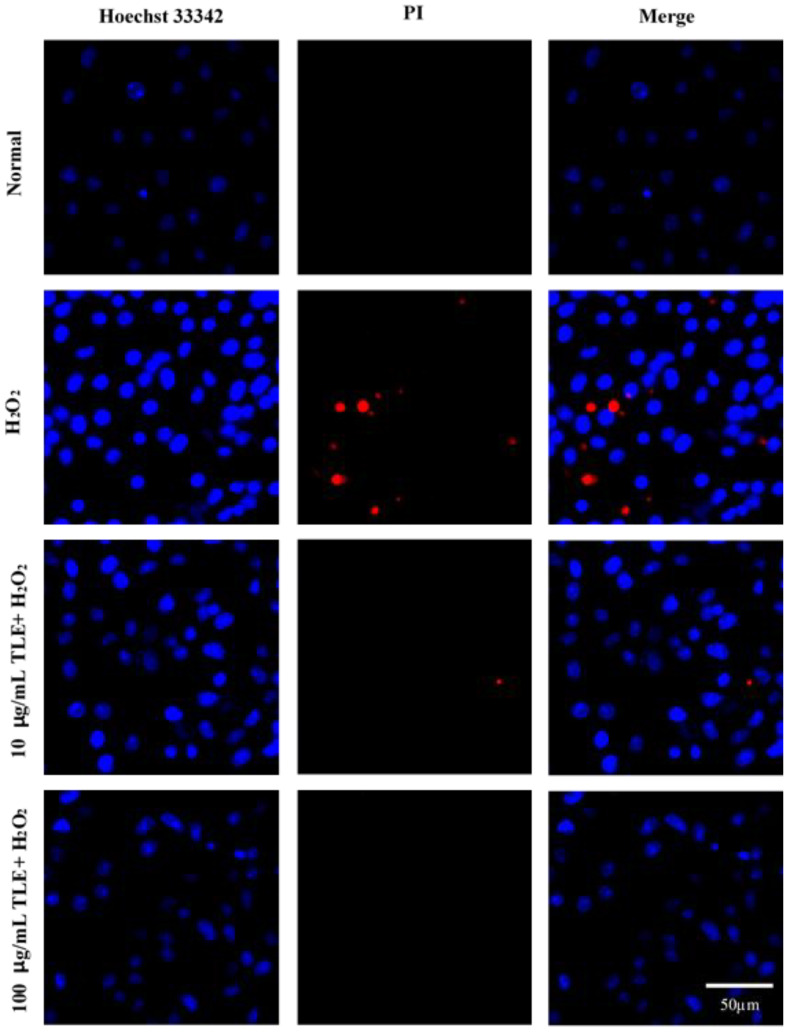
Fluorescence microscopy images of dual-stained Vero cells treated with TLE and 600 μM H_2_O_2_. H_2_O_2_ was treated to Vero cells, which were pretreated with 10 μg/mL and 100 μg/mL TLE. Nuclei were stained with Hoechst 33342 (blue) and dead cells were stained with propidium iodide (PI) (red). Scale bar: 50 μm. TLE, turmeric leaves extract; PI, Propidium iodide.

**Figure 6 antioxidants-10-00112-f006:**
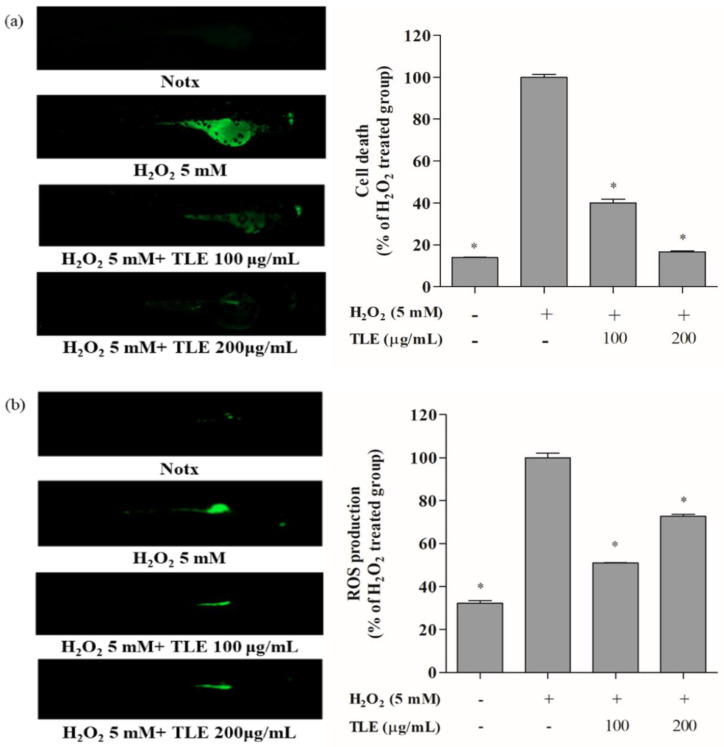
Cell death, ROS production, and lipid peroxidation in zebrafish. Zebrafish embryos were pre-treated with TLE (50, 100, and 200 μg/mL) and/or treated with H_2_O_2_ (5 mM). Imaging of (**a**) cell death, (**b**) ROS production, and (**c**) lipid peroxidation was performed using fluorescence microscopy. The fluorescence intensity of zebrafish was quantified using ImageJ software. The data were measured in triplicate and expressed as the mean ± SE. * *p* < 0.05 compared with the H_2_O_2_-treated group.

## Data Availability

Data are available from corresponding authors on request.

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
