# Peer review of "Antioxidant Effects of Turmeric Leaf Extract against Hydrogen Peroxide-Induced Oxidative Stress In Vitro in Vero Cells and In Vivo in Zebrafish"

_antioxidants, 2021, doi:10.3390/antiox10010112_

Round 1

Reviewer 1 Report

The manuscript ID: antioxidants-1074319 - title: “Antioxidant Effects of Turmeric Leaf Extract against Hydrogen Peroxide-induced Oxidative Stress in vitro in Vero cells and in vivo in Zebrafish” , is a research article describing the antioxidant properties  of extract of Turmeric leaves (Curcuma longa L. leaves; TL) both in vitro and in vivo.

The manuscript is professionally written and extensively deals with many aspects of oxidative stress, even though there are several typos and grammatical errors throughout the text.

COMMENT

Authors used the Turmeric Leaf Extract that contains many natural phenols (curcuminoids). They report (Lane 13- 170- 174) that the TLE has antioxidant properties, including DPPH, ABTS and radical scavenging activity but in the Method section of the manuscript the assays are not described.

I suggest adding the methods used and explaining as the values of these TLE activities were obtained (lane 170).

Authors assert that their study provided the first evidence demonstrating that TLE has scavenging properties in vitro and in vivo, but they never measured the antioxidant activity of the extract.

The antioxidant protective effect of TLE could result in the triggering of signaling pathways that have not been considered.

Lane 290- Please, add some more information about the leaf age and the leaf harvest season.

Lane 293- The aqueous leaf extract, after filtration, was stored at -20℃.  In the manuscript Authors used TLE in μg/mL and in Section 2.1, the HPLC-MS analysis of flavonoids is reported in ng/mg.  

Has the aqueous extract been dried?

Authors are requested to be more precise about the concentration used and report the TLE values correctly and in agreement with respect to what they described in the Methods.

Lane 344. Indicate the wavelengths used for the observation nuclear staining under a fluorescence microscope

Section 4.4. Cell culture- This section needs to be expanded and more information on cell growing conditions needs to be given.

References must be correct, and some the authors' names are incorrect. Furthermore, all references lack the DOI

Author Response

Thank you for the comments and kind advice. We sincerely responded to all of your comments. Please see the attachment and review the responses.

Sincerely,

Sera Kim

Reviewer 2 Report

Manuscript number Antioxidants-1074319

entitled: Antioxidant Effects of Turmeric Leaf Extract against Hydrogen Peroxide-induced Oxidative Stress in vitro in Vero cells and in vivo in Zebrafish

This is valuable and a well-conducted scientific study, done thoroughly and expressed concisely. Therefore, the manuscript is suitable for Antioxidants after considering the below comments:

  1. Please add space between, e.g., 37space°C (page 10, line 311; and in other places).
  2. Please provide chemical structures of main chemicals which contain the turmeric leaves., close to Fig. 1
  3. From the present version of the manuscript, we do not know which product of the turmeric leaves is responsible for biological activity. Do we know the mechanism of biological activity of the main chemicals that contain the turmeric leaves?
  4. The Authors provide information about the organic part of the turmeric leaves. Do we know the turmeric leaves' inorganic contents, especially transition metals like Fe? (see the well-known Fenton reaction).

Author Response

(The authors gave the same response as above.)

Round 2

Reviewer 1 Report

The Authors took into account the comments of the reviewers.

The manuscript has been improved and the present form of the paper can be accepted.

I believe that the present version of the paper is suitable for publishing.